# Enhancing Optical and Thermal Stability of Blue-Emitting Perovskite Nanocrystals through Surface Passivation with Sulfonate or Sulfonic Acid Ligands

**DOI:** 10.3390/nano14121049

**Published:** 2024-06-18

**Authors:** Shu-Han Huang, Sheng-Hsiung Yang, Wen-Cheng Tsai, Hsu-Cheng Hsu

**Affiliations:** 1Institute of Lighting and Energy Photonics, College of Photonics, National Yang Ming Chiao Tung University, Tainan 711010, Taiwan; heaven142857@gmail.com; 2Department of Photonics, National Cheng Kung University, Tainan 701401, Taiwan; l76114348@gs.ncku.edu.tw (W.-C.T.); hsuhc@mail.ncku.edu.tw (H.-C.H.); 3Program on Key Materials, Academy of Innovative Semiconductor and Sustainable Manufacturing, National Cheng Kung University, Tainan 701401, Taiwan

**Keywords:** perovskite nanocrystals, surface passivation, sulfonate ligands, photoluminescence quantum yield, thermal stability

## Abstract

This study aims to enhance the optical and thermal properties of cesium-based perovskite nanocrystals (NCs) through surface passivation with organic sulfonate (or sulfonic acid) ligands. Four different phenylated ligands, including sodium β-styrenesulfonate (SbSS), sodium benzenesulfonate (SBS), sodium *p*-toluenesulfonate (SPTS), and 4-dodecylbenzenesulfonic acid (DBSA), were employed to modify blue-emitting CsPbBr_1.5_Cl_1.5_ perovskite NCs, resulting in improved size uniformity and surface functionalization. Transmission electron microscopy and X-ray photoelectron spectroscopy confirmed the successful anchoring of sulfonate or sulfonic acid ligands on the surface of perovskite NCs. Moreover, the photoluminescence quantum yield increased from 32% of the original perovskite NCs to 63% of the SPTS-modified ones due to effective surface passivation. Time-resolved photoluminescence decay measurements revealed extended PL lifetimes for ligand-modified NCs, indicative of reduced nonradiative recombination. Thermal stability studies demonstrated that the SPTS-modified NCs retained nearly 80% of the initial PL intensity when heated at 60 °C for 10 min, surpassing the performance of the original NCs. These findings emphasize the optical and thermal stability enhancement of cesium-based perovskite NCs through surface passivation with suitable sulfonate ligands.

## 1. Introduction

All-inorganic cesium-based perovskite nanocrystals (NCs) have garnered widespread attention since the first report was published in 2015 [1] due to their high color purity, tunable bandgap, defect tolerance, and high photoluminescence quantum yields (PLQYs) [2,3,4,5]. Perovskite NCs can be deposited into compact and defect-free thin films through solution processes, and can be utilized in several fields such as perovskite light-emitting diodes (PeLEDs), lasers, photodetectors, and perovskite solar cells [6,7,8,9]. Various methods have been proposed to prepare perovskite NCs, including hot injection approach, low-temperature synthesis, solvothermal synthesis, ultrasonication treatment, and microwave-assisted synthesis [10,11,12,13,14,15]. Among them, the hot injection approach is the most popular way to yield NCs with uniform size distributions and high crystallinity.

During the synthesis of perovskite NCs, oleylamine (OAm) and oleic acid (OA) are commonly used ligands which serve a dual purpose. First, a large quantity of ligands is required to provide sufficient surface passivation, thereby eliminating surface defects and promoting the PLQY of perovskite NCs. Second, using long-chain ligands can assure the good dispersion of perovskite NCs in organic solvents. However, the poor conductivity of organic compounds like OAm and OA leads to the formation of an insulating layer outside perovskite NCs when excessive amounts of ligands are present, hindering charge injection into the perovskite cores and reducing the performance of PeLEDs [16]. Song, Zeng, and their coworkers obtained a fiftyfold external quantum efficiency (EQE) improvement of PeLEDs by using a solvent mixture consisting of hexane and ethyl acetate to remove excessive surface ligands and optimize charge injection [17]. By applying two cycles of mixed solvent treatment, a good balance between carrier injection and surface passivation was achieved. Quaternary ammonium salts like didodecyldimethylammonium bromide are frequently adopted during the synthesis of perovskite NCs, with elevated quantum yields up to 95% and enhanced stability compared to those capped with OA and OAm [18].

Through relentless efforts in the past decade, the PLQYs of green and red-emitting perovskite NCs have approached 96–100% [19,20]. PeLEDs based on green-emitting CsbBr_3_ NCs or red-emitting CsPbI_3_ NCs with EQEs exceeding 20% were reported [21,22]. However, the PLQYs of blue-emitting perovskite NCs are relatively low. The surface-to-volume ratio of NCs rapidly increases as the size decreases, leading to a large number in surface defects that decrease PLQY. Achieving high PLQY in the blue and deep blue emission spectrum is extremely challenging. The study of mixed halide perovskites with the general formula of APbCl_x_Br_y_I_3−x−y_ (A = Cs, MA, FA) is a hot topic in the research of pure blue-light emitters [23,24,25]. By simply adjusting the ratio of Cl, Br, and I anions, precise blue emission can be finely tuned. Regrettably, in mixed halide perovskites with high chlorine composition, it is commonly accepted that the existence of chlorine vacancies generates deep trap states within the bandgap that severely stuck carriers and suppress the charge recombination [26,27,28]. Additionally, halide ions are reported to segregate spontaneously through defect-assisted ion migration channels, resulting in poor spectral stability [29,30]. Several methodologies have been documented to tackle these challenges. Park et al. utilized potassium thiocyanate (KSCN) to suppress defect states in CsPbBr_x_Cl_3−x_ NCs [31]. Both K^+^ and SCN^−^ ions can efficiently fill halide vacancies and uncoordinated halide sites. The PLQY of the KSCN-introduced perovskite NCs was significantly augmented from 34.0% (pristine NCs) to 74.1%. Luo et al. proposed an ultrafast thermodynamic control strategy by adding liquid nitrogen into the reaction mixture to freeze the superior crystal lattices of CsPbBr_x_Cl_3−x_ NCs at high temperatures [32]. The final perovskite NCs possessed more uniform crystalline size and improved PLQY compared to the untreated NCs. Wang et al. devised a novel two-step hot-injection approach to fill Cl vacancies by incorporating Br precursors like ZnBr_2_ and CuBr_2_ via efficient anion exchange, yielding highly blue-emissive CsPb(Cl/Br)_3_ NCs [33]. Cohen et al. reported zwitterion-functionalized poly(isopropyl methacrylate) containing sulfonate anions and quaternary ammonium cations to stabilize green- to infrared-emitting perovskite NCs, maintaining their PL quantum yields in a composite film state even after one year of ambient storage [34]. Zhang et al. used cesium–dodecyl-benzene sulfonic acid to passivate CsPbBr_3_ perovskite quantum dots with a high PLQY up to 100% and good color stability [35]. It can be seen that organic ligands with sulfonate anions or sulfonic acid groups can effectively passivate lead halide perovskite NCs.

In this work, we present a series of surface-passivated CsPbBr_1.5_Cl_1.5_ perovskite NCs by incorporating phenylated sulfonate (or sulfonic acid) ligands into the Pb precursor solution. The original NCs exhibit numerous uncoordinated Cl^−^/Br^−^ vacancies on the perovskite surface, leading to significant nonradiative recombination and low device performance [36,37]. The sulfonate (or sulfonic acid) ligands can occupy these halide vacancies through S=O, thus effectively passivating the surface defects [38]. The chemical structures of the proposed ligands, including sodium β-styrenesulfonate (SbSS), sodium benzenesulfonate (SBS), sodium p-toluenesulfonate (SPTS), and 4-dodecylbenzenesulfonic acid (DBSA) are depicted in Figure 1a. Meanwhile, Figure 1b illustrates the surface passivation mechanism of sulfonate (or sulfonic acid) ligands onto perovskite NCs. Experimental results reveal that the perovskite NCs passivated by sulfonate (or sulfonic acid) ligands showed a significant enhancement in PL intensity and stability compared to the original perovskite NCs.

## 2. Materials and Methods

### 2.1. Materials

Cesium carbonate (Cs_2_CO_3_, purity 99.9%), lead(II) bromide (PbBr_2_, purity 99%), lead(II) chloride, (PbCl_2_, purity 99%), and (OAm (purity 99%) were bought from Alfa Aesar (Haverhill, MA, USA). 1-Octadecene (ODE, purity 90%), (SbSS (purity 95%), (DBSA (purity 95%), and (STS (purity 90.0%) were purchased from TCI (Tokyo, Japan). (SBS (purity 98%) was bought form Thermo Scientific (Waltham, MA, USA). (OA (purity 90%) was bought from Sigma-Aldrich (St. Louis, MO, USA). Hexane and ethyl acetate (EA) were received from Alfa Aesar and ECHO (Miaoli, Taiwan), respectively. All chemicals were used without further purification. For the fabrication of PeLEDs, indium tin oxide (ITO) glass substrates (15 Ω/square) were purchased from Aimcore Technology Co., Ltd. (Hsinchu, Taiwan). Poly(3,4-ethylenedioxythiophene):polystyrene sulfonate (PEDOT:PSS) aqueous solution (Clevios PVP AI 4083) was received from Heraeus Precious Metals GmbH & Co. KG (Hanau, Germany). Poly[(9,9-dioctylfluorenyl-2,7-diyl)-co-(4,4′-(N-(4-sec-butylphenyl)diphenylamine)] (TFB) was bought from Lumtec (New Taipei City, Taiwan). 1,3,5-Tris(1-phenyl-1H-benzimidazol-2-yl)benzene (TPBi) was purchased from Shine Material Technology (Kaohsiung, Taiwan).

### 2.2. Preparation of Cs-Oleate

Cs_2_CO_3_ (0.407 g, 1.25 mmol), OA (1.25 mL, 3.96 mmol), and 20 mL of ODE were loaded into a three-necked flask, evacuated, and heated to 120 °C for 1 h for degassing. Nitrogen was pumped into the reaction flask for 10 min, followed by degassing for 20 min. Then, nitrogen was pumped into the flask again, and the reaction mixture was heated at 160 °C for 20 min to obtain the Cs-oleate solution, which was kept at 30 °C for storage. The solution was heated to 100 °C before use.

### 2.3. Synthesis of CsPbBr_1.5_Cl_1.5_ NCs

PbCl_2_ (0.0523 g, 0.188 mmol), PbBr_2_ (0.069 g, 0.188 mmol), OA (0.6 mL, 1.9 mmol), OAm (1 mL, 3.04 mmol), and 10 mL of ODE were placed in a three-necked flask, evacuated, and heated to 120 °C for 1 h for degassing. Nitrogen was pumped into the reaction flask for 10 min, followed by degassing for 20 min. Then, nitrogen was pumped into the flask again, and the reaction mixture was heated at 180 °C for 20 min. Afterward, 0.8 mL of Cs-oleate solution was injected into the above reaction mixture using a syringe. After 5 s, the reaction flask was quickly immersed in an ice-water bath. The crude solution was centrifuged at 8500 rpm for 10 min, and the supernatant was discarded. Then, the precipitate was dispersed in 2 mL of hexane, followed by adding 8 mL of EA and centrifuging at 8500 rpm for 10 min. The precipitate was collected and re-dispersed in 2 mL of hexane.

For those perovskite NCs containing sulfonate (or sulfonic acid) ligands, 0.0188 mol of different ligand was solely added into the PbCl_2_/PbBr_2_ solution. The synthetic procedure was the same as that of the original perovskite NCs.

### 2.4. Fabrication of Devices

The devices with the regular configuration of ITO/PEDOT:PSS/TFB/perovskite NCs/TPBi/LiF/Al were fabricated. The ITO-coated glass substrates were cleaned sequentially with detergent, deionized water, acetone, and isopropanol under ultrasonication for 20 min each, followed by nitrogen purging and oxygen plasma treatment for 3 min. The PEDOT:PSS solution was spin-cast onto the ITO substrate at 4000 rpm for 40 s and annealed at 150 °C for 15 min. The TFB solution (8 mg/mL in chlorobenzene) was deposited on top of PEDOT:PSS by spin-coating at 3000 rpm for 40 s and annealed at 120 °C for 20 min. Original or modified perovskite NCs solutions were spin-cast on the TFB layer at 2000 rpm for 60 s and annealed at 80 °C for 10 min. Finally, TPBi, LiF, and Al electrodes were thermally evaporated with thicknesses of 40, 1, and 100 nm, respectively, under a base pressure of 8 × 10^−6^ Torr. The active area of each device was 1 mm^2^ for performance evaluation.

### 2.5. Characterization

The photoluminescence (PL) and absorption spectra of samples in solution state were recorded with a Princeton Instruments Acton 2150 spectrophotometer equipped with a xenon lamp (ABET Technologies LS 150, Milford, CT, USA) as the light source. The Fourier transform infrared (FT-IR) spectra were measured using a Thermo Scientific Nicolet iS-10 spectrometer for the analysis of functional groups. The ^1^H nuclear magnetic resonance (NMR) spectra were recorded on a Bruker Avance III HD 600 MHz NMR spectrometer. Deuterated dimethyl sulfoxide (DMSO-D_6_) was used as the d-solvent. The morphology and size of perovskite NCs were examined with an ultrahigh resolution JEOL JEM-2100F (Tokyo, Japan) transmission electron microscope (TEM). The accelerating voltage used for TEM, high resolution (HR-TEM), and high-angle annular dark-field scanning TEM (HAADF-STEM) observation was 100, 600, and 1200 kV, respectively. The size distribution of perovskite NCs was statistically determined using ImageJ software version 1.54d. The X-ray photoelectron spectroscopy (XPS) measurements were performed using a Thermo K-Alpha XPS instrument for the elemental composition analysis of the perovskite NCs. The detection angle was set to 60°.The X-ray diffraction (XRD) patterns of samples were performed with a Bruker D8 SSS instrument. The time-resolved photoluminescence (TR-PL) decay measurement was performed on the time-correlated single photon counting (TCSPC) system with a PicoHarp 300 (PicoQuant, Berlin, Germany) acquisition unit, which was assembled with a 377 nm pulsed diode laser (PicoQuant model: LDH-P-C-375B). The emitted light was collected by using a spectrometer (Horiba iHR-320, Kyoto, Japan) equipped with a photomultiplier tube (PMT) (South-Port Co., Taipei, Taiwan) The PL emission was collected from a 50× objective lens in a Horiba iHR320 spectrometer equipped with a liquid nitrogen-cooled charge-coupled device array detector. For the evaluation of PeLEDs, the performance of light-emitting devices was recorded using an Agilent 4155C semiconductor parameter analyzer (Santa Clara, CA, USA) and an Ocean Optics USB2000+ spectrometer (Orlando, FL, USA).

## 3. Results and Discussion

### 3.1. TEM and Elemental Distribution Analysis

Figure 2a–e shows TEM images of the original and sulfonate (or sulfonic acid)-modified perovskite NCs, revealing cubic crystalline structures. Especially, the DBSA-containing NCs exhibit a relatively uniform size among all samples. The insets in Figure 2 display the HRTEM images of perovskite NCs without and with various sulfonate or sulfonic acid ligands, all demonstrating a lattice spacing of 0.41 nm, which corresponds to the (110) plane of the cubic perovskite structure [39]. The size distribution histogram is integrated in Figure 2f. It is seen that the crystal size varies with the introduction of different organic ligands. The average crystal size of the original perovskite NCs is approximately 11.6 ± 1 nm. By incorporating SbSS, SBS, and SPTS ligands, the average crystal sizes of NCs gradually increase to 12.4 ± 1.4, 13.9 ± 1.97, and 17.4 ± 1.7 nm, respectively. In contrast, the DBSA-containing NCs have a reduced crystal size of 10.1 ± 1.3 nm. In addition, Appendix A shows the HAADF-STEM images and elemental mapping of perovskite NCs without and with various sulfonate or sulfonic acid ligands. In the detection area, the distribution of S, Cl, Br, Cs, and Pb elements is consistent with the corresponding HAADF-STEM image, indicating that the phenylated sulfonate or sulfonic acid ligands were successfully attached onto the surface of perovskite NCs.

### 3.2. XRD Analysis

The XRD patterns of the original and sulfonate (or sulfonic acid)-modified perovskite NCs were meticulously analyzed to unravel the crystalline structure of perovskite materials, as shown in Figure 3. Several diffraction peaks at 2θ = 15.5°, 21.5°, 30.3°, 35.3°, 37.4°, and 45.5° are observed, corresponding to the prominent (100), (110), (200), (210), (211), and (220) planes of the cubic perovskite structure [40,41]. These patterns can be distinctly identified as the cubic phase by comparing them with the standard cubic CsPbCl_3_ (PDF#18-0366) and CsPbBr_3_ (PDF#18-0364). A Scherrer analysis of the XRD peaks was applied to realize the size distribution of perovskite NCs. According to the calculated results, the crystal sizes are 16.7, 17.6, 18.35, 17.89, and 15.9 nm for the original and SbSS-, SPTS-, SBS-, DBSA-modified perovskite NCs, respectively, which are slightly larger than the measured results from the TEM observation.

### 3.3. XPS Analysis

We employed the XPS technique to investigate the interaction between perovskite NCs and different sulfonate (or sulfonic acid) ligands. Figure 4a shows the Pb 4f_5/2_ and Pb 4f_7/2_ core-level signals of the original NCs at 142.9 and 138.0 eV, respectively [42]. After modification with sulfonate (or sulfonic acid) ligands, the Pb 4f_5/2_ and Pb 4f_7/2_ signals of the modified perovskite NCs show negative shifts to 142.53–142.88 and 137.73–137.98 eV, respectively. The decrease in binding energy suggests a reduction in the charge of Pb^2+^ cations in perovskite NCs, indicating that the lone electron pairs on oxygen atoms of sulfonate groups (–SO^3−^) enter the empty 6p orbitals of Pb^2+^ for coordination bonding [43], particularly the SbSS ligand, which brings the most significant shift on XPS peak positions. In Figure 4b, the Br 3d_5/2_ and 3d_3/2_ peaks of the original NCs are located at 69.2 and 68.1 eV, respectively [44]. In Figure 4c, the Cl 2p_1/2_ and 2p_3/2_ peaks of the original NCs are found at 199.18 and 197.63 eV, respectively. Similar to the Pb 4f signals, the Br 3d and Cl 2p XPS spectra of perovskite NCs are shifted to lower binding energies after modifying with different sulfonate (or sulfonic acid) ligands. As sulfonate (or sulfonic acid) ligands approach Pb^2+^ cations to form coordination interaction, the surrounding Cl^−^ or Br^−^ ions in perovskite NCs would be disturbed. There were no significant changes in the Cs 3d signals of the original and modified perovskite NCs in Figure 4d. In Appendix A, the S 2p core-level spectra of perovskite NCs without and with sulfonate (or sulfonic acid) ligands are presented. These spectra also confirm the presence of sulfonate (or sulfonic acid) ligands on perovskite NCs.

### 3.4. FT-IR Analysis

To further study the existence of sulfonate (or sulfonic acid) ligands on the perovskite NCs, a series of FT-IR characteristic bands in Figure 5 were analyzed. The absorption bands at 2925 and 2852 cm^−1^ are attributed to C–H asymmetric stretching of the CH_2_ group [45,46]. About the three bands observed at 2958, 1450, and 1380 cm^−1^; the first two characteristic bands are assigned to the asymmetric stretching of the terminal –CH_3_ group, and the last one corresponds to the symmetric bending of the –CH_3_ group. The absorption bands observed at 1712 and 1240 cm^−1^ are attributed to the C=O group of OA and the C–N stretching of OAm, respectively [47,48]. Turning to characterization of phenylated ligands, the two bands at 1609 and 1493 cm^−1^ correspond to the C=C stretching of the benzene ring. The three sulfonates SbSS, SBS, and SPTS present two absorption bands at 1190 and 1046 cm^−1^, indicating asymmetric and symmetric S=O stretching modes, respectively [45]. As for DBSA, the S=O stretching bands of sulfonic acid group appear at 1180 and 1040 cm^−1^ [45]. In addition, the =CH in-plane bending of the benzene ring was observed at 1013 cm^−1^, while the out-of-plane bending was found at 832 cm^−1^. Herein, sulfonate (or sulfonic acid) ligands are certainly incorporated onto the perovskite NCs according to FT-IR characterization.

### 3.5. NMR Analysis

This study conducted ^1^H NMR experiments to verify whether sulfonate (or sulfonic acid) ligands were bound to the surface of the perovskite NCs. Figure 6 displays ^1^H NMR spectra of the original and modified perovskites with different ligands (SbSS, SBS, DBSA, and SPTS). Taking SbSS as an example, the multiple signals at around 7.2–7.5 ppm are assigned to the protons on the benzene ring, and the doublet of doublets at 6.8–6.9 ppm is derived from vinyl protons. These proton signals can be observed for the SbSS-modified perovskite NCs, indicative of the existence of SbSS on the perovskite. Similarly, the aromatic protons of SBS, DBSA, and SPTS are observed at δ = 7.1–7.6 ppm, which are also observable for the corresponding modified perovskite NCs. To further determine the relative proportion of surface ligands, the characteristic peaks of hydrogen atoms on the benzene ring from sulfonate (or sulfonic acid) ligands are located at 7.2–7.5 ppm. The vinyl proton signals of OA/OAm are found at 5.3 ppm. The neighboring CH_2_ groups near the NH_3_^+^ (from OAm) and COO^−^ (from OA) are found at 2.7 and 1.95 ppm, respectively. Based on the peak areas of the aromatic protons, vinyl protons, and CH_2_ groups from OA/OAm, we calculated the relative proportions of OA:OAm = 4.02:1 for the original perovskite NCs, OA:OAm:SPTS = 5.47:2.48:1 for the SPTS-modified perovskite NCs, OA:OAm:DBSA = 5.97:3.88:1 for the DBSA-modified perovskite NCs, and OA:OAm:SBS = 2.76:1.54:1 for the SBS-modified perovskite NCs. As for the SbSS-modified perovskite NCs, the relative proportion of the surface ligands SbSS to OA/OAm is calculated to be 7.54:1. The relative molar ratio of OA to OAm is undistinguishable due to severe proton signal overlaps.

### 3.6. Optical Measurements

The UV-vis absorption spectra of the original perovskite NCs and those passivated by sulfonate (or sulfonic acid) ligands are displayed in Appendix A. The corresponding Tauc plot is shown in Figure 7a, revealing the bandgap energy (E_g_) of 2.73–2.78 eV. The E_g_ value slightly decreased when passivated by sulfonate ligands, whereas it increased when using DBSA for surface passivation. The PL emission spectra of the original and modified perovskite NCs are depicted in Figure 7b. The PL maximum of the original perovskite NCs was observed at 448 nm, which was shifted to 450, 451, 452, and 440 nm for the SbSS, SBS, SPTS, and DBSA-modified perovskite NCs, respectively. All synthesized perovskite NCs showed blue emission with full width at half maximum (FWHM) values of 27–29 nm. Similar to UV-vis absorption, the PL emission maxima of the perovskite NCs passivated by sulfonate ligands (SbSS, SBS, and SPTS) are red-shifted, while the sulfonic acid (DBSA)-modified NCs show a blue-shifted PL emission. The shifts in UV-vis absorption and PL emission are attributed to quantum confinement effects, meaning that optical properties of NCs are determined by their nanocrystalline sizes [49]. As can be seen from TEM observations, the crystal sizes of the sulfonate ligands (SbSS, SBS, and SPTS)-modified NCs are larger than that of the original NCs, resulting in red-shifted absorption and emission behaviors. On the contrary, the DBSA-modified NCs show a blue shift in absorption and emission spectra due to the smaller crystal size. From Figure 7b, it is also seen that the PL intensity of the SbSS-, SBS-, and SPTS-modified perovskite NCs is stronger than that of the original NCs, while the DBSA-modified perovskite has the lowest emission intensity. The PLQY of the original perovskite NCs was measured to be 32%, and it elevated to 48%, 38%, and even 63% for the SbSS-, SBS-, and SPTS-modified perovskite NCs, respectively. This emission enhancement is attributed to the reduction in surface defects by introducing sulfonate ligands, thereby enhancing radiative recombination [50]. However, the PLQY of the perovskite modified with the DBSA ligand decreased to 18%. This could be due to the smaller crystal size of NCs when using the DBSA ligand, resulting in increased surface area and more surface traps that induced carrier quenching to lower PLQY.

To verify the radiative recombination process of perovskite NCs with and without sulfonate (or sulfonic acid) ligands, we performed TR-PL decay experiments, and the corresponding TR-PL decay curves are shown in Figure 7c. The TR-PL decay curves were fitted with a three-component exponential decay model [51]. The detailed lifetime parameters for TR-PL decay measurements are listed in Appendix A. The fast decay (τ_1_) originates from the non-radiative quenching of carriers, and the slow decay (τ_2_) comes from radiative recombination of electron hole pairs. The third decay (τ_3_) refers to an additional decay process such as radiative recombination of free charge carriers. The average PL lifetimes of the original and SbSS-, SBS-, SPTS-, and DBSA-modified perovskite NCs were calculated to be 17.0, 32.6, 23.9, 43.6, and 7.1 ns, respectively. The results of TR-PL decay measurements align well with the trend of PLQY, showing that the SbSS, SBS, and SPTS-modified perovskite NCs exhibit longer PL lifetime, especially the SPTS-modified NCs. To realize the thermal stability of NCs, the purified NC dispersions were heated at various temperatures from 30 to 100 °C for PL measurements. The heating temperature was raised every 10 °C, which was held for 10 min, followed by measuring their PL intensity. The PL spectra of the original and modified perovskite NCs with sulfonate (or sulfonic acid) ligands at different heating temperatures are presented in Appendix A, and the temperature-dependent PL intensity evolution is illustrated in Figure 7d. The PL intensity of the original perovskite NCs decreased by 50% when it was heated to 60 °C. In contrast, the PL intensity of the SPTS-modified NCs retained nearly 80% of its initial value when heated to the same temperature, and it dropped to 50% at a temperature exceeding 100 °C. It is also seen that the thermal stability of those NCs modified with DBSA, SbSS, and SBS ligands is better than that of the original perovskite NCs at the same heating temperature, demonstrating the effective enhancement of thermal stability for modified perovskite NCs. Through TEM observation, we realize that the SPTS-modified perovskite NCs have the most uniform size and intact crystalline shape, as shown in Figure 2d,f. The XRD analysis also reveals that it has the highest crystallinity from those diffraction peaks along the (110), (200), and (210) planes. We conclude that the SPTS-modified perovskite NCs possess the best optical properties and stability among all NCs. Figure 7e shows the snapshot of the original and modified perovskite NC solutions under UV light (375 nm) exposure in ambient atmospheric conditions. It is clearly seen that the SPTS- and SBS-modified NC solutions look brighter than other solutions. We fabricated PeLEDs with the configuration of ITO/PEDOT:PSS/TFB/perovskite NCs/TPBi/LiF/Al for device evaluation, using the original or modified perovskite NCs as the active layer. The brightness–voltage and current-efficiency–current-density characteristics of PeLEDs are provided in Appendix A. Preliminary results show that the device based on the SPTS-modified perovskite NCs exhibited a maximum luminance of 350 cd m^−2^ and the highest current efficiency (η_max_) of 0.1 cd A^−1^, which are clearly better than other devices (for example, L_max_ = 160.7 cd m^−2^ and η_max_ = 0.025 cd A^−1^ for the control device). It is evident that introducing the sulfonate (or sulfonic acid) ligands can improve the quality of perovskite NCs and the performance of PeLEDs. Future work will be focused on the optimization of blue-emitting PeLEDs using the original and modified perovskite NCs as the active layer.

## 4. Conclusions

In this work, we successfully synthesized a series of blue-emitting perovskite NCs with sulfonate (or sulfonic acid) ligands via the hot injection method. TEM observation confirmed that the crystal size of perovskite NCs was finely tuned upon the introduction of different ligands. XPS results revealed that coordination interaction was formed between the sulfonate group and Pb^2+^ cations, while FT-IR and NMR experiments verified the incorporation of sulfonate (or sulfonic acid) ligands onto NCs. PL and TR-PL decay measurements demonstrated blue emission for all perovskite NCs and prolonged carrier lifetime for those capped with SPTS, SBS, or SbSS ligands. Thermal stability studies revealed that the SPTS-modified NCs retained nearly 80% of its initial PL intensity at 60 °C, surpassing the performance of the original NCs. Overall, using appropriate sulfonate ligands as the surface passivant can mitigate surface defects and enhance the quality of perovskite NCs toward future optoelectronic applications.

## Figures and Tables

**Figure 1 nanomaterials-14-01049-f001:**
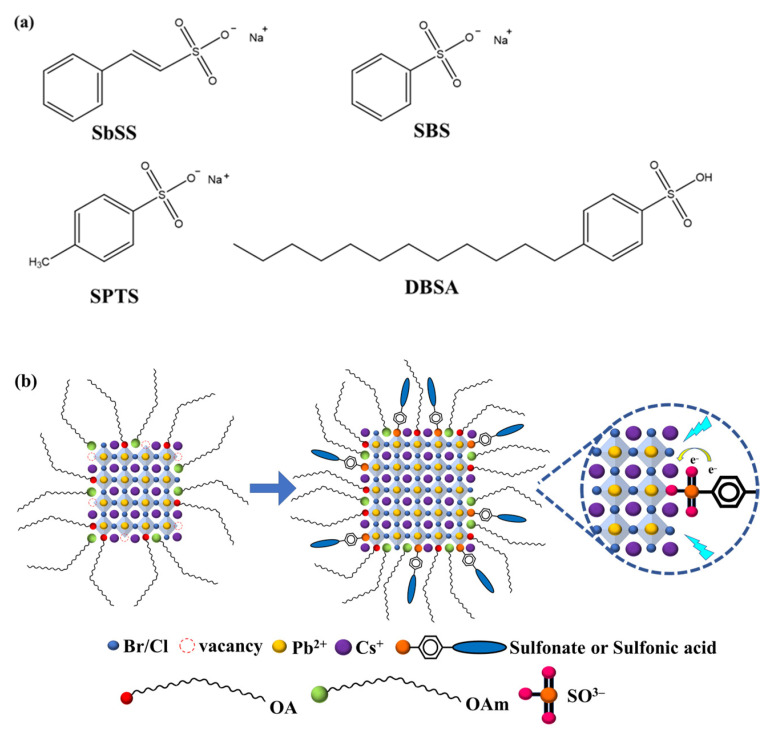
(**a**) Chemical structures of SbSS, SBS, SPTS, and DBSA; (**b**) schematic illustration of the sulfonate (or sulfonic acid) ligands onto the perovskite NCs.

**Figure 2 nanomaterials-14-01049-f002:**
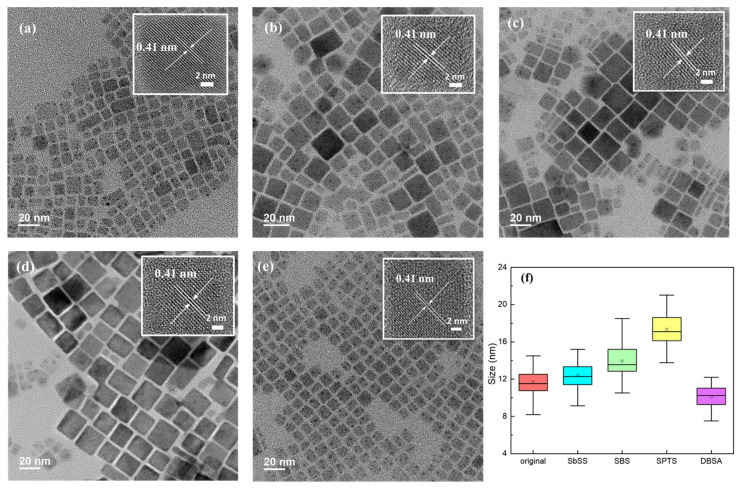
TEM and HR-TEM images of the (**a**) original and (**b**) SbSS-, (**c**) SBS-, (**d**) SPTS-, (**e**) DBSA-modified perovskite NCs; (**f**) size distribution of perovskite NCs with various organic ligands.

**Figure 3 nanomaterials-14-01049-f003:**
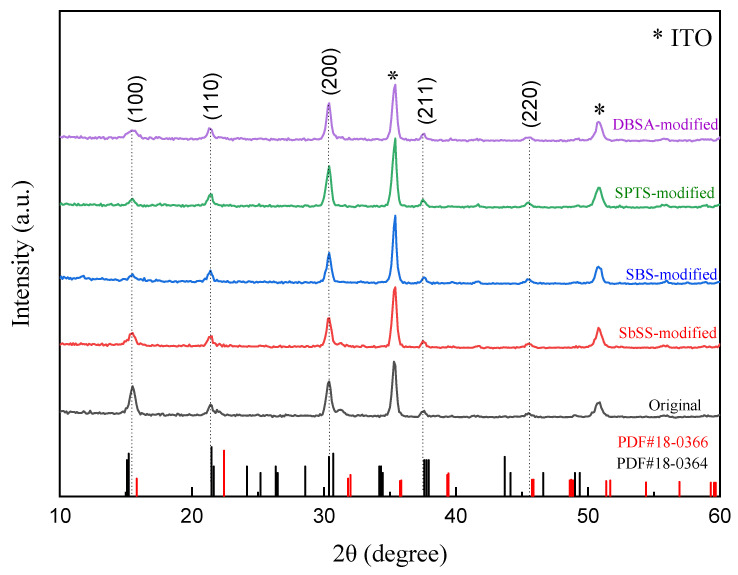
XRD patterns of the original and modified perovskite NCs.

**Figure 4 nanomaterials-14-01049-f004:**
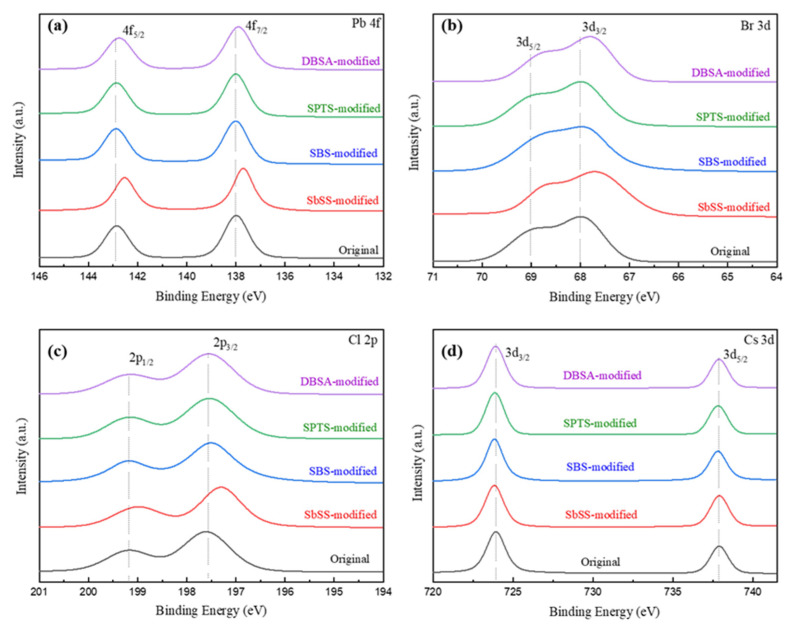
(**a**) Pb 4f, (**b**) Br 3d, (**c**) Cl 2p, and (**d**) Cs 3d XPS spectra of perovskite NCs without and with sulfonate (or sulfonic acid) ligands.

**Figure 5 nanomaterials-14-01049-f005:**
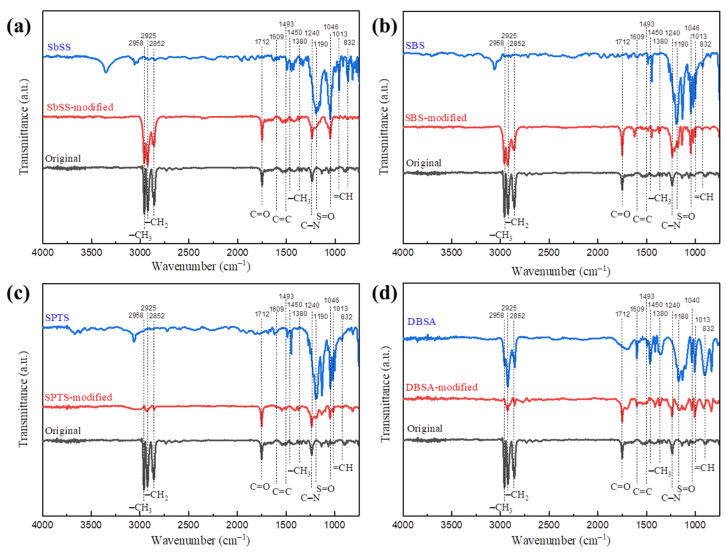
FT-IR spectra of the original perovskite, sulfonate (or sulfonic acid) ligands, and modified perovskites with (**a**) SbSS, (**b**) SBS, (**c**) SPTS, and (**d**) DBSA.

**Figure 6 nanomaterials-14-01049-f006:**
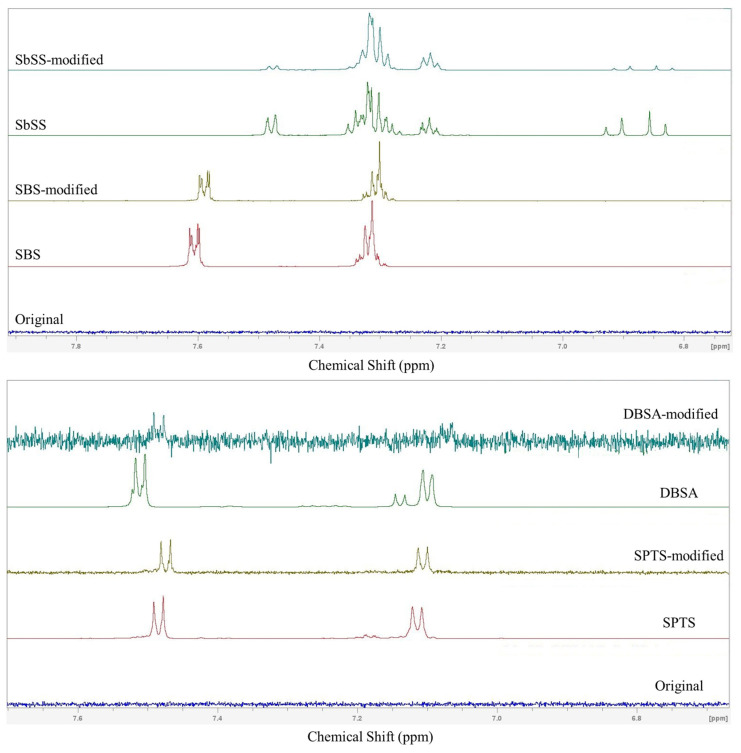
^1^H NMR spectra of the ligands (SbSS, SBS, DBSA, and SPTS), the original perovskite, and ligand-modified perovskites.

**Figure 7 nanomaterials-14-01049-f007:**
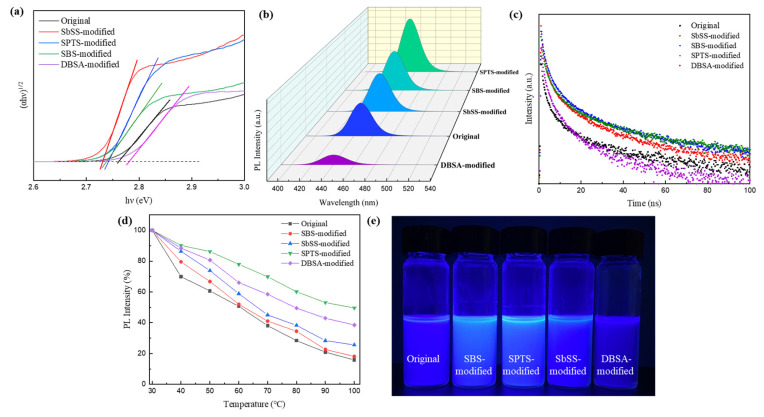
(**a**) Tauc plot, (**b**) PL emission spectra, (**c**) TR-PL decay curves, (**d**) temperature-dependent PL intensity evolution, and (**e**) the solutions of the perovskite NCs exposed to UV light (375 nm).

## Data Availability

Data is contained within the article.

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
