# Peer review of "Enhancing Optical and Thermal Stability of Blue-Emitting Perovskite Nanocrystals through Surface Passivation with Sulfonate or Sulfonic Acid Ligands"

_nanomaterials, 2024, doi:10.3390/nano14121049_

Round 1
Reviewer 1 Report
Comments and Suggestions for Authors
Review attached

Moderate changes required
Author Response
1. In general, I would suggest checking the English throughout the manuscript, since sometimes there are typos or sentences that could be written better. Some examples: a) Line 48 “prohibiting” could be replaced by “hindering” b) Line 68 “the existence of chlorine vacancies is common that generates” could be replaced by “it is commonly accepted that the existence of chlorine vacancies generates” c) Line 69 “to severely stuck” -> “that severely stuck” d) Line 91 “Meanwhile” is incorrect in this context. Use e.g. “Instead” or similar.
Response: We thank for the reviewer’s suggestion. We have consulted a professional English editing office to polish the whole text. The mentioned typos and sentences have been amended as follows. a) the word “prohibiting” is replaced by “hindering” in line 48; b) the sentence is re-written as “it is commonly accepted that the existence of chlorine vacancies generates deep trap states…” c) the description is amended as “… within the band gap that severely stuck carriers…” d) The word “Meanwhile” is replaced by “On the contrary” in the revised manuscript.
2. It appears that two acronyms “SβSS” and “SbSS” are used but they appear to refer to the same thing (see e.g. line 15 and line 106). Please fix this inconsistency.
Response: Thanks for pointing this out. The acronym “SbSS” is unified in the revised manuscript. We are sorry for the mistake.
3. Line 295-296 “The TR-PL decay curves were fitted with a three-component exponential decay model”. Some more details could be given, e.g, the actual equation used to fit the curves, and the (three?) parameters obtained and how they relate to the lifetime; this can be reported in main or in the Supp Info, but I think it should be discussed somewhere.
Response: Thanks for the reviewer’s comment. The equation and carrier lifetime parameters are summarized in Table S1 in the Supporting Information. The fast decay (τ1) originates from the non-radiative quenching of carriers and the slow decay (τ2) comes from radiative recombination of electron-hole pairs. The third decay (τ3) refers to an additional decay process such as radiative recombination of free charge carriers. The above description is added in the 3.6 Optical measurements section in the revised manuscript.
Reviewer 2 Report
Comments and Suggestions for Authors
The manuscript presents a promising procedure for the passivation of PVK materials.
The organization and structure of the work are fine and the data are sufficient. However, the discussion is not enough and it should be modified. In the current form, there is not enough comparison to previous works and other surface passivation methods in terms of optical, morphological, chemical, and crystallographical points of view.
Some comments as examples (and not limited to):
1. In materials and methods: some more details of the characterizations are required:
1a. number of measurements/each condition,
1b. Detection angle in XPS?
1c. TEM and SEM: What was the voltage?
2. Line 172: "...determined by ImageJ software....". This part is more appropriate for materials and methods.
3. What does the shift of Pb4f in XPS (Figure 4 a) indicate? How does it correlate with other studies?
Comments on the Quality of English Language
Moderate English grammar required.
Author Response
- We thank for the reviewer’s comment and details of the characterizations are provided as follows. 1a) We performed two measurements for XPS, NMR, XRD experiments, and three measurements for TEM, FT-IR, absorption, PL, and TR-PL decay experiments; 1b) In the XPS measurements, the detection angle was set to 60°; 1c) The accelerating voltage used for TEM, HR-TEM, and HAADF-STEM observation was 100, 600, and 1200 kV, respectively. The above details 1b) and 1c) are added in the 2.5 Characterization section in the revised manuscript. We did not carry out SEM experiment and therefore the description about SEM instrument is deleted.
- Thanks for the reviewer’s suggestion. The sentence “The size distribution of perovskite NCs was statistically determined by ImageJ software” is moved to the 2.5 Characterization section.
- As stated in the 3.3 XPS analysis section, the Pb 4f signals of the modified perovskite NCs show negative shifts from 138–142.9 eV to 137–142.5 eV after ligands modification. The decrease in binding energy suggests a reduction in the charge of Pb2+ cations in perovskite NCs, indicating that the lone electron pairs on oxygen atoms of sulfonate groups (–SO3–) enter the empty 6p orbitals of Pb2+ for coordination bonding. Here we also cited the ref [43] for comparison, showing the same tendency in binding energy change of Pb 4f signal.
- Thanks for the reviewer’s suggestion. We have consulted a professional English editing office to polish the text. Please check the revised manuscript with the “Track Change” function.
Reviewer 3 Report
Comments and Suggestions for Authors
The author explored using sulfate or sulfonate group to in-situ passivate lead halide perovsktie nanocrystals during synthesis. There are several points to be addressed before being published:
1) In the experimental section, please includes the solvents to run the NMRs, and I suggest the author use the NMR to cacualte the ratios among ammonnium, carboxylate and sulfonic ligands;
2) For size analysis, in addition to TEM, I encourage to use Scherrer analysis of the XRD to get the size distrition on an ensemble level for comparision and reference;
3) The authors should explain the different shift between the SbSS-modified NCs and the sulfonic and sulfate based ligands. The current statement claims the shift in XPS indicate Pb2+ passivation, which suggests that SPTS and SBS doesn't passivate the NCs. This is contradicting to the whole paper;
4) Please include the recent papers that use sulfonic or sulfate groups for lead halide perovskite NC passivation:
https://doi.org/10.1016/j.cej.2023.145213
https://doi.org/10.1021/acs.chemmater.1c00902
Author Response
- We thank for the reviewer’s suggestion. Deuterated dimethyl sulfoxide (DMSO-D6) was used as the d-solvent to run the NMR experiment. The above sentence is added in the 2.5 Characterization section. To further determine the relative proportion of surface ligands, the characteristic peaks of hydrogen atoms on the benzene ring from sulfonate (or sulfonic acid) ligands are located at 7.2–7.5 ppm. The vinyl proton signals of OA/OAm are found at 5.3 ppm. The neighboring CH2 groups near the NH3+ (from OAm) and COO– (from OA) are found at 2.7 and 1.95 ppm, respectively. Based on the peak areas of the aromatic protons, vinyl protons, and CH2 groups from OA/OAm, we calculated the relative proportions of OA:OAm = 4.02:1 for the original perovskite NCs, OA:OAm:SPTS = 5.47:2.48:1 for the SPTS-modified perovskite NCs, OA:OAm:DBSA = 5.97:3.88:1 for the DBSA-modified perovskite NCs, and OA:OAm:SBS = 2.76:1.54:1 for the SBS-modified perovskite NCs. As for the SbSS-modified perovskite NCs, the relative proportion of the surface ligands SβSS to OA/OAm is calculated to be 7.54:1. The relative molar ratio of OA to OAm is undistinguishable due to severe proton signal overlaps. The above description is added in the 3.5 NMR analysis section in the revised manuscript.
- Thanks for the reviewer’s suggestion. The Scherrer analysis of the XRD peaks was applied to realize the size distribution of perovskite NCs. According to the calculated results, the crystal sizes are 16.7, 17.6, 18.35, 17.89, and 15.9 nm for the original and SbSS-, SPTS-, SBS-, DBSA-modified perovskite NCs, respectively, which are slightly larger than the measuring results from TEM observation. The above description is added in the 3.2 XRD analysis section in the revised manuscript.
- Thanks for pointing this out. As mentioned in the 3.3 XPS analysis section, after modification with sulfonate (or sulfonic acid) ligands, the Pb 4f signals of the modified perovskite NCs show negative shifts to 137.73–142.88 eV compared to the original NCs. In fact, the Pb2+ XPS signals for the SPTS- and SBS-modified perovskite NCs is also shifted to lower binding energies. However, this shift is not as pronounced as it is in the SbSS- and DBSA-modified perovskite NCs, as shown in Figure 4a. The expression “SPTS and SBS doesn’t passivate the NCs” was not stated in the whole paper.
- Thanks for the reviewer’s suggestion. After reading the two suggested papers, we found them useful to the readers and cited as ref [34] and [35] in the revised manuscript. Cohen et al. reported zwitterion-functionalized poly(isopropyl methacrylate) containing sulfonate anions and quaternary ammonium cations to stabilize green- to infrared-emitting perovskite NCs, maintaining their PL quantum yields in composite film state even after one year of ambient storage. Zhang et al. used a cesium-dodecyl benzene sulfonic acid to passivate CsPbBr3 perovskite quantum dots with high PLQY up to 100% and good color stability. It can be seen that organic ligands with sulfonate anions or sulfonic acid groups can effectively passivate lead halide perovskite NCs. The above description is added in the third paragraph in the 1. Introduction section.
Round 2
Reviewer 2 Report
Comments and Suggestions for Authors
The authors addressed the issues and the manuscript can be considered for publication.
Comments on the Quality of English LanguageThe authors addressed the issues and the manuscript can be considered for publication.